# Challenges and costs of donor screening for fecal microbiota transplantations

Mèlanie V. Bénard[1][☯]*, Clara M. A. de Bruijn[1,2,3][☯], Aline C. Fenneman[4,5], Koen Wortelboer[5], Judith Zeevenhoven[2], Bente Rethans[1], Hilde J. Herrema[5], Tom van Gool[6], Max Nieuwdorp[5], Marc A. Benninga[2][‡], Cyriel Y. Ponsioen[1][‡]

**1** Department of Gastroenterology and Hepatology, Amsterdam Gastroenterology Endocrinology Metabolism (AGEM), Amsterdam UMC, University of Amsterdam, Amsterdam, The Netherlands, **2** Emma Children's Hospital, Amsterdam UMC, University of Amsterdam, Pediatric Gastroenterology, Hepatology and Nutrition, Amsterdam, The Netherlands, **3** Amsterdam Reproduction & Development Research Institute, Amsterdam UMC, Emma Children's Hospital, Amsterdam, The Netherlands, **4** Department of Endocrinology and Metabolism, Amsterdam Gastroenterology Endocrinology Metabolism (AGEM), Amsterdam UMC, University of Amsterdam, Amsterdam, The Netherlands, **5** Department of Clinical and Experimental Vascular Medicine, Amsterdam Cardiovascular Sciences (ACS), Amsterdam UMC, University of Amsterdam, Amsterdam, The Netherlands, **6** Section Clinical Parasitology, Department of Medical Microbiology, Amsterdam University Medical Centers, Amsterdam Medical Center, Amsterdam, The Netherlands

☯ These authors contributed equally to this work.
‡ MAB and CYP also contributed equally to this work.
* m.v.benard@amsterdamumc.nl

**Data Availability Statement:** The data supporting the findings of this study are available within the paper and its Supporting information files.

## Abstract

### Background

The increasing interest to perform and investigate the efficacy of fecal microbiota transplantation (FMT) has generated an urge for feasible donor screening. We report our experience with stool donor recruitment, screening, follow-up, and associated costs in the context of clinical FMT trials.

### Methods

Potential stool donors, aged between 18–65 years, underwent a stepwise screening process starting with an extensive questionnaire followed by feces and blood investigations. When eligible, donors were rescreened for MDROs and SARS-CoV-2 every 60-days, and full rescreening every 4–6 months. The costs to find and retain a stool donor were calculated.

### Results

From January 2018 to August 2021, 393 potential donors underwent prescreening, of which 202 (51.4%) did not proceed primarily due to loss to follow-up, medication use, or logistic reasons (e.g. COVID-19 measures). 191 potential donors filled in the questionnaire, of which 43 (22.5%) were excluded. The remaining 148 candidates underwent parasitology screening: 91 (61.5%) were excluded, mostly due to *Dientamoeba* fragilis and/or high amounts of *Blastocystis* spp. After additional feces investigations 18/57 (31.6%) potential donors were excluded (mainly for presence of Helicobacter Pylori and ESBL-producing

**Funding:** MN is supported by a ZONMW VICI grant 2020 [09150182010020]. There were no other specific grant for this research from any funding agency in the public, commercial or not-for-profit sectors. The funders (ZONMW) had no role in study design, data collection and analysis, decision to publish, or preparation of the manuscript.

**Competing interests:** he authors have declared that no competing interests exist.

organisms). One donor failed serum testing. Overall, 38 out of 393 (10%) potential donors were enrolled. The median participation time of active stool donors was 13 months. To recruit 38 stool donors, €64.112 was spent.

## Conclusion

Recruitment of stool donors for FMT is challenging. In our Dutch cohort, failed eligibility of potential donors was often caused by the presence of the protozoa *Dientamoeba* fragilis and *Blastocystis* spp.. The exclusion of potential donors that carry these protozoa, especially *Blastocystis* spp., is questionable and deserves reconsideration. High-quality donor screening is associated with substantial costs.

## Introduction

Fecal microbiota transplantation (FMT) is defined as the infusion of feces from healthy individuals into diseased recipients. FMT is thought to be effective because it has the potential to restore a recipient's distorted microbiota, by introducing a new and diverse microbiome associated with a healthy state to normalize microbiota composition and function. In daily practice, FMT is a widely accepted and highly effective treatment for recurrent *Clostridioides difficile* infection (CDI) [1, 2]. Over the past couple of years, evidence is growing for the application of FMT as a treatment for other diseases, such as inflammatory bowel disease (IBD) [3], irritable bowel syndrome (IBS) [4], obesity and related metabolic diseases [5], acute graft-versus-host disease [6], and autism spectrum disorder [7]. The interest in FMT increased tremendously recently, with more than 357 registered ongoing clinical trials worldwide at the time of writing [8–10].

This increasing interest in FMT has generated an urge for feasible donor screening programs to secure an ongoing supply of healthy stool donors. Enrolled donors need to fulfill strict safety criteria, which are continuously adjusted to new insights [11]. For example, due to the current COVID-19 pandemic, additional screening procedures to assess COVID-19 symptoms before donation and regular testing for SARS-CoV-2 RNA are needed [12, 13]. In addition, measures to reduce the risk of transmitting multi-drug resistant organisms (MDROs) via FMT were advised earlier by the United States Food and Drug Administration (FDA) after two immunocompromised adults developed invasive infections with extended-spectrum beta-lactamase (ESBL)-producing *Escherichia coli* [11]. Although international recommendations on donor screenings exist [14], stool donor selection processes in practice are highly heterogeneous, and standardized procedures are lacking [9]. Experience from clinical practice indicates that finding a safe, eligible stool donor is complicated. Previous studies performed in the USA, Canada, Hong Kong, and Denmark have shown variable donor acceptance rates ranging between 0.8–31% [15–21]. Challenges in donor screening comprise initial donor recruitment and prolonged donor eligibility. A major disadvantage of the extensive screening procedures is the high associated costs [15], leading to an economic burden for patient care and research departments. Therefore, more insights into donor screening programs and accompanying costs are warranted to optimize and further standardize donor screening procedures.

At present, limited data is published on FMT donor screening and associated costs within the context of clinical trials. In recent years, one of the largest University hospitals in the Netherlands −the Amsterdam UMC− has conducted four randomized controlled clinical FMT trials: the FAIS [22], IMITHOT and PIMMS trials have evaluated the efficacy of multiple donor

FMTs using fresh fecal material in respectively IBS (in adolescents), subclinical autoimmune hypothyroidism and metabolic syndrome, whereas the TURN2-trial is evaluating the efficacy of frozen fecal suspensions in active ulcerative colitis. To perform these trials, a pool of healthy stool donors who were able to provide regular stool donations was established. The donor screening was performed according to a predefined standardized screening protocol. With the current study, we aim to describe the process of recruiting and screening stool donors, evaluate the follow-up of eligible donors, and report the associated costs in the context of clinical FMT trials in a Dutch tertiary University hospital.

## Methods

### Donor recruitment

In this retrospective observational cohort–study, potential healthy fecal donors were recruited through advertisements via posters, announcements in the hospital magazine and intranet network (employee website), and word-of-mouth advertising among staff at the Amsterdam UMC (location AMC). The Amsterdam UMC, location AMC, is a University hospital with over 7000 employees and 2300 healthcare student placements. Potential donors were invited to participate in the FAIS, IMITHOT, PIMMS, and/or TURN2-trial and oral and written information about the study aims, donation process, and screening requirements were provided. Clinical trials registration numbers are NCT03074227, NL7931, NL8289, and NL7770, respectively. All trials were approved by the Medical Ethics Research Committee of the Amsterdam UMC, the Netherlands. Written, signed and dated informed consent forms were obtained separately for each study as participation in multiple trials was optional. Financial compensation was offered for qualified fecal donors, with reimbursements ranging between €10–50 per donation plus additional travel expenses, depending on the trial. No compensation was offered during the screening phase.

### Population and screening procedure

The study population consisted of non-smoking adults, aged 18 – 65 years (except for the TURN2 trial, in which the age ranged between 18–54 years), and with a body mass index between 18 – 25 kg/m.$^2$ No specific diet restrictions were required. After informed consent was signed, potential donors were thoroughly screened based on the screenings protocol of the Netherlands Donor Feces Bank (NDFB) [23], a Dutch stool bank that supplies FMT for the treatment of CDI in the Netherlands since 2016. Before accepting a donor, a rigorous screening was performed as shown in Table 1. The screening started with an extensive questionnaire regarding risk factors for infectious diseases and factors potentially perturbing the intestinal microbiota. When potential donors passed the screening questionnaire, they subsequently underwent elaborate fecal and blood laboratory testing in a stepwise approach (Table 2). First, stool samples -collected in a plastic stool container- were screened for parasites presence by a combination of PCR and direct microscopy (Dual Feces Test). Next, feces samples were tested for pathogenic bacteria and viruses, multi-drug resistant organisms and calprotectin. Subsequently, routine biochemical analysis of blood was performed, followed by serological testing for pathogenic viruses, bacteria, and parasites. Once qualified as fecal donors, rescreening of active fecal donors was performed regularly to reduce the risk of transmission of infectious diseases as much as possible. In line with FDA recommendations [11], screening for MDROs (fecal culture) and molecular stool testing on SARS-CoV-2 was performed every 60 days. Frozen FMT material (TURN2 trial) remained quarantined until successful complete rescreening, performed every four months. Complete rescreening was executed every six months when fresh FMT was used (other trials). During the trials, the study staff were in regular contact with

**Table 1. Exclusion criteria donor recruitment.**

| Risk of infectious agent |
|---|
| Active hepatitis A, B-, C- or E-virus infection or known exposure within recent 12 months |
| Acute infection with *Cytomegalovirus* (CMV) or *Epstein-Barr virus* (EBV) |
| An extensive travel behavior |
| Higher risk of colonization with multidrug resistant organisms including: |
| • *Health care workers with direct patient contact* |
| • *Persons who have recently been hospitalized or discharged from long term care facilities* |
| • *Persons who regularly attend outpatient medical or surgical clinics* |
| • *Persons who have recently engaged in medical tourism* |
| History or current use of (IV) drugs |
| Individual working with animals[a] |
| Positive blood tests for the presence of: HIV, HTLV, *Treponema pallidum*, *Strongyloides stercoralis* |
| Positive fecal test for MDROs, pathogenic bacteria, viruses and parasites as listed in Table 2 |
| Previous reception of blood products (<12 months) or recent needle-stick accident (<6 months)[a] |
| Tattoo or body piercing placement within last 6 months |
| Unsafe sex practice (assessed with standardized questionnaire) |
| **Gastrointestinal comorbidities** |
| A positive history/clinical evidence (e.g. elevated fecal calprotectin) for inflammatory bowel disease, including Crohn's disease or ulcerative colitis |
| A positive history/clinical evidence for other gastrointestinal diseases, including chronic diarrhea or chronic constipation |
| Abnormal bowel motions, abdominal complaints or symptoms indicative of irritable bowel syndrome |
| **Factors affecting intestinal microbiota composition** |
| Antibiotic treatment in the past 12 weeks[b] |
| History of or present known malignant disease and/or patients who are receiving systemic anti-neoplastic agents |
| History of cholecystectomy |
| History of treatment with growth factors |
| Patients receiving immunosuppressive medications and/or a positive history/clinical evidence for autoimmune disease including: |
| • *Type 1 Diabetes Mellitus* |
| • *Hashimoto's hypothyroidism* |
| • *Graves' hyperthyroidism* |
| • *Rheumatoid arthritis* |
| • *Celiac disease* |
| Recent (gastrointestinal) infection within last 6 months[c] |
| Smoking |
| Use of any medication including PPI, except contraceptives and over the counter medication |
| Use of pre- and probiotics in the past 12 weeks[a] |
| **Other conditions** |
| Abnormal liver function[d]: ASAT >40 U/L, ALAT >45 U/L, AF >120 U/L, GGT >60 U/L, bilirubin >17µmol/L |
| Abnormal renal function[d]: creatinine >110 µmol/l, urea >8,2 mmol/l |
| Alcohol abuse (>3 units/day) |
| Chronic pain syndromes (e.g., fibromyalgia)[c] |
| Impaired immunity[d]: CRP >5 mg/L, hemoglobin <8,5 mmol/L, MCV 80–100 fL, leukocytes 4,0–10,5 x10$^9$/L, thrombocytes 150–400 x10$^9$/L |
| Known chronic neurological/neurodegenerative disease (e.g., Parkinson's disease, multiple sclerosis) |
| Known psychiatric disease (i.e., depression, schizophrenia, autism, Asperger's syndrome) |
| Known risk of Creutzfeldt Jacob's disease |
| Major relevant allergies (e.g., food allergy, multiple allergies) |
| Presence of diabetes mellitus type 1 and 2 or hypertension[d] |
| Presence of chronic low-grade inflammation or metabolic syndrome (NCEP criteria)[e] |

[a] Not included in screening protocol of FAIS and TURN2-trial;

[b] For the TURN2-trial the exclusion criteria included antibiotic treatment in the past 4 weeks;

[c] Additional exclusion criteria FAIS trial;

[d] Not included in screening protocol of TURN2-trial;

[e] Additional exclusion criteria PIMMS trial

Abbreviations: AF, alkaline phosphatase; ALAT, alanine aminotransferase; ASAT, aspartate aminotransferase; GGT, gamma-glutamyl transferase; CRP, c-reactive protein; HIV, human immunodeficiency virus; HTLV, human T-lymphotropic virus; MCV, mean corpuscular volume; MDROs, multidrug-resistant organisms, NCEP, National Cholesterol Education Programs; PPI, proton pump inhibitors.

**Table 2. Specification of donor screening and associated costs.**

| Feces screening | | € | |
|---|---|---|---|
| **Calprotectin[a] (ELISA)** | | ***20,-*** | |
| ***Bacteria*** *(PCR or stool antigen detection[b])* | | ***150,-*** | |
| *Clostridium difficile* | *Salmonella spp.* | | |
| *Helicobacter pylori* | Shiga toxin-producing *Escherichia coli* (STEC) | | |
| Pathogenic *Campylobacter spp.* | *Shigella spp.* | | |
| *Plesiomonas shigelloides* | *Yersinia enterocolitica* | | |
| ***Multidrug resistant organisms*** *(culture)* | | ***150,-*** | |
| Carbapenem-resistant *Enterobacteriaceae* (CRE) | Multidrug-resistant Gram-negatives (MRGN) 3 | | |
| ESBL-producing *Enterobactereacceae* | MRGN 4 | | |
| Methicillin-resistant *Staphylococcus aureus* (MRSA) | Vancomycin-resistant *Enterococcus* (VRE) | | |
| ***Viruses*** *(PCR)* | | ***125,-*** | |
| Adenovirus non-41/41 | Norovirus Type I and II | 45 | |
| Adenovirus type 40/41 | Parechovirus | | |
| Astrovirus | Rotavirus | | |
| Enterovirus | Sapovirus | | |
| Severe acute respiratory syndrome coronavirus 2 (SARS-CoV-2) | | 45 | |
| Hepatitis E virus | | 35 | |
| ***Parasites*** *(PCR and/or microscopic evaluation)* | | ***212,-*** | |
| *Blastocystis* spp.[c] | *Entamoeba moshkovskii[d]* | | |
| *Cryptosporidium* spp. | *Entamoeba polecki* [d] | | |
| *Cyclospora* | *Giardia lamblia* | | |
| *Dientamoeba fragilis* | *Iodamoeba bütschlii[d]* | | |
| *Endolimax nana[d]* | *Isospora spp.* | | |
| *Entamoeba coli[d]* | Larvae[c] | | |
| *Entamoeba dispar[d]* | *Microsporidium spp.* | | |
| *Entamoeba gingivalis[d]* | Parasitic worm eggs[c] | | |
| *Entamoeba hartmanni[d]* | Protozoan Cysts and Oocysts[c] | | |
| *Entamoeba histolytica* | | | |
| **Serum screening** | | | |
| ***Hematology[a]*** | | ***44,-*** | |
| Alanine aminotransferase (ALAT) | Complete Blood Count (CBC) | | |
| Alkaline phosphatase (AF) | C-reactive protein (CRP) | | |
| Aspartate aminotransferase (ASAT) | Estimated Glomerular Filtration Rate (EGFR) | | |
| Bilirubin | Kreatinin | | |
| Gamma-glutamyl transferase (GGT) | Ureum | | |
| ***Bacteria*** *(ELISA)* | | ***8,-*** | |
| *Treponema pallidum* | | | |
| ***Viruses[d]*** *(CLIA or PCR)* | | ***Serology: 119,-*** | ***PCR: 293,-*** |
| *Cytomegalovirus* (CMV) | | 36,- | 35,- |
| *Epstein-Barr Virus* (EBV) | | 25,- | |
| *Hepatitis A virus[a]* | | 15,- | |
| *Hepatitis B virus* | | 10,- | 67,- |
| *Hepatitis C virus* | | 11,- | 77,- |
| *Human immunodeficiency viruses* (HIV) | | 11,- | 63,- |
| *Human T-lymphotropic virus Type I and II (HTLV)* | | 11,- | |
| ***Parasites*** *(ELISA)* | | ***18,-*** | |

(*Continued*)

**Table 2.** (Continued)

| Feces screening | | € |
|---|---|---|
| *Strongyloides stercoralis* | | 18 |

[a] Not included in screening protocol of TURN2-trial

[b] All bacteria were detected with the use of PCR, with exception of Helicobacter pylori were ELISA was used

[c] Microscopic evaluation, exclusion of donor only if high amounts *Blastocystis* spp. are seen, defined as 'moderate' or 'many' [26]

[d] Presence of only one non-pathogenic parasite is acceptable

Abbreviations: ELISA, quantitative enzyme-linked immunosorbent assay; CLIA, chemi-luminescence immunoassay.

the active stool donors, especially before each donation. If there were any concerns about symptoms or risk factor exposure of the fecal donor, donation was suspended and an additional rescreening was performed. In addition, since the outbreak of coronavirus pandemic in 2019 (COVID-19) questions to assess the risk on SARS-CoV-2 infection were asked, including the presence of fever, cough, sore throat, dyspnea, anosmia or ageusia, or close contact to subjects with suspected or proven infection. Independent of SARS-CoV-2 vaccination status, in case of any suspicion on COVID-19 infection, nasopharyngeal swab and reverse transcription polymerase chain reaction (RT-PCR) were performed and the potential donor was temporarily excluded. During the screening and rescreening process, all positive laboratory tests were discussed with the (potential) donor and counselling was provided accordingly. Qualified fecal donors were matched to patients based on gender (with exception of the TURN2-trial) and cytomegalovirus (CMV)/ Epstein–Barr virus (EBV) status. Donors of the TURN2-trial were additionally selected on a putatively favorable microbiota profile based on results from a previous TURN1 trial, including high alpha-diversity and high predicted butyrate production [24, 25].

## Data and statistics

Data were collected from January 2018 to August 2021. To date, donor recruitment is still carried out for the IMITHOT and TURN2-trial. Data were collected in the Electronic Data Capture system Castor EDC. Descriptive statistics were used to summarize variables. Normally distributed continuous data are expressed as mean (SD). Not normally distributed continuous data are presented as median (IQR). Categorical data are displayed as frequencies (percentages). Data were analyzed using IBM SPSS Statistics for Windows, Version 26.0 (Armonk, NY: IBM Corp).

## Results

### Initial donor screening

From January 2018 to August 2021, a total of 393 potential donors underwent prescreening. A flowchart of donor screening is presented in Fig 1. The main causes for failing prescreening were lost to follow-up (N = 97), logistics problems (N = 35, *e.g.*, working from home as a result of national COVID-19 measures), occupation as a health care worker with direct patient contact (N = 23), and the use of medication, including pre- and probiotics (N = 19). Eventually, only half of the initial respondents signed informed consent and continued the screening procedure (N = 195). After consenting, four individuals did not respond to further communication and were lost to follow-up. All other potential donors filled in the online screening questionnaire (N = 191). Based on 191 completed questionnaires, 43 individuals (23%) were

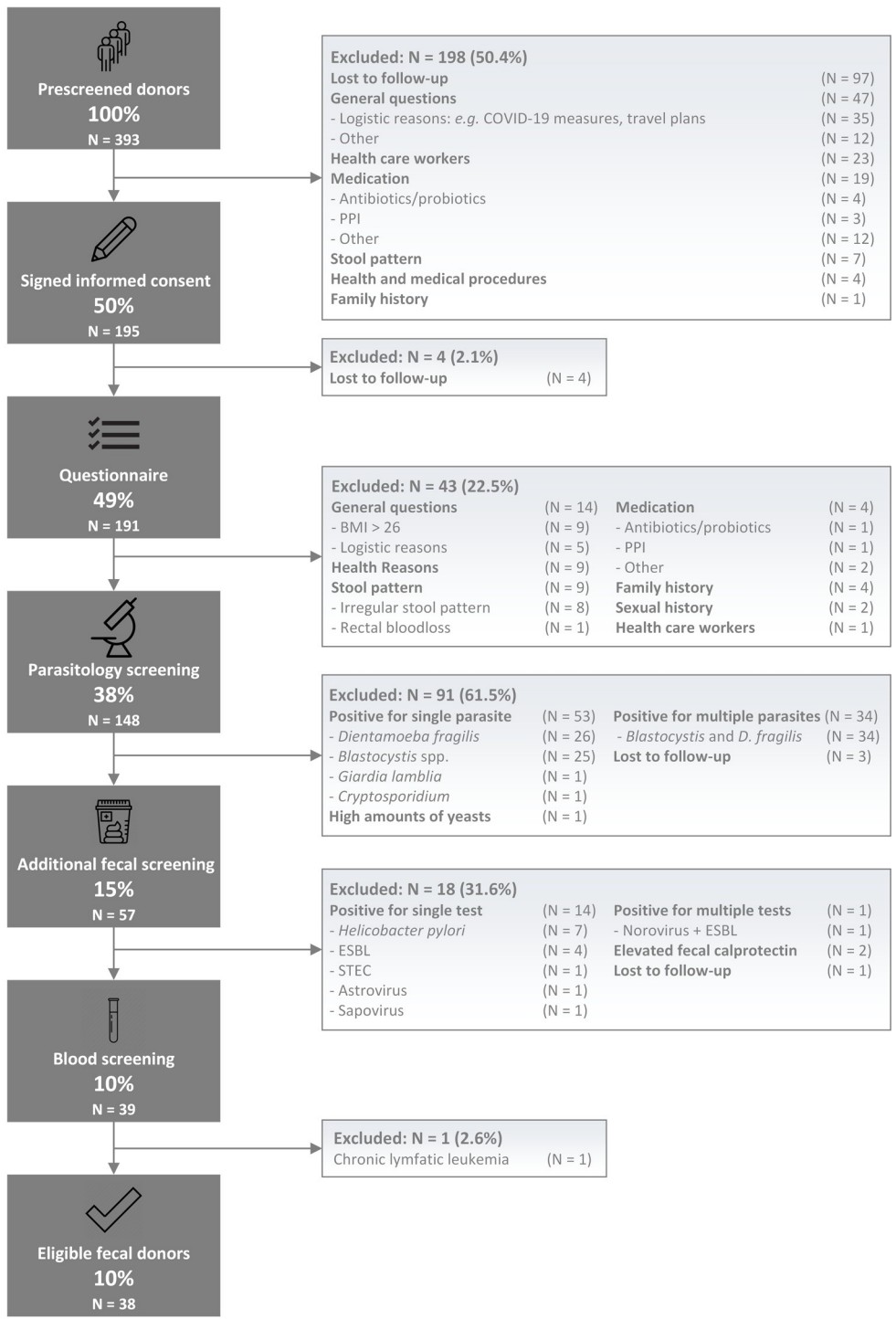

**Fig 1. Flow diagram of donor screening outcomes.**

excluded for various reasons (Fig 1). Hereafter, 148 potential donors remained and sent in fecal samples for parasitology screening. This screening step resulted in the largest relative loss of potential donors, with positive test results in 91 out of 148 samples (61%). Potential donors tested most frequently positive for *Dientamoeba fragilis* (N = 26, 29%), microscopic

quantification of 'moderate' or 'many' *Blastocystis spp.* (N = 25, 27%), or a combination of both (N = 34, 37%). Asymptomatic infestation with *Giardia Lamblia* and *Cryptosporidium* resulted in the exclusion of two additional donors. One donor was dismissed from further screening steps because remarkably high amounts of yeasts were noticed during microscopy evaluation of the stool. Next, 57 potential donors continued screening and delivered stool samples for biochemical, bacterial, and viral analysis. Eighteen out of 57 individuals (32%) failed these stool tests: 7 had Helicobacter pylori, 4 an ESBL-producing strain of E. coli, 1 individual had a Shiga toxin-producing *E.coli* (STEC), 2 potential donors tested positive for a pathogenic virus (astrovirus, sapovirus) and one individual tested positive for multiple tests (norovirus plus an ESBL). Two additional potential donors were excluded due to elevated fecal calprotectin levels (79 and 87 ug/g). The penultimate screening step consisted of blood analysis and resulted in the exclusion of only one individual who had remarkably high levels of lymphocytes and was later diagnosed with chronic lymphatic leukemia. Serum screening for the presence of Hepatitis B and C, HIV, recent infection of CMV and EBV, *Strongyloides*, and *Treponema pallidum* didn't result in any positive tests. In the end, only 38 of the initial 393 individuals (10%) could be enrolled as fecal donors.

## Eligible fecal donors

A flowchart of the follow-up of eligible donors is presented in Fig 2. The median age of the 38 eligible fecal donors was 28 years (IQR: 25–31.5 years), and 14 donors (36.8%) were male. Eligible donors had a healthy weight with a median BMI of 22.5 kg/m$^2$ (IQR: 20.3–24.0 kg/m$^2$). Twenty-four of the 38 eligible fecal donors (63.2%) donated at least one time, further referred to as 'active donors'. The other 14 'non-active' donors could not be matched to a patient due to their microbiota profile (TURN2-trial), gender and/or CMV/EBV status, and therefore did

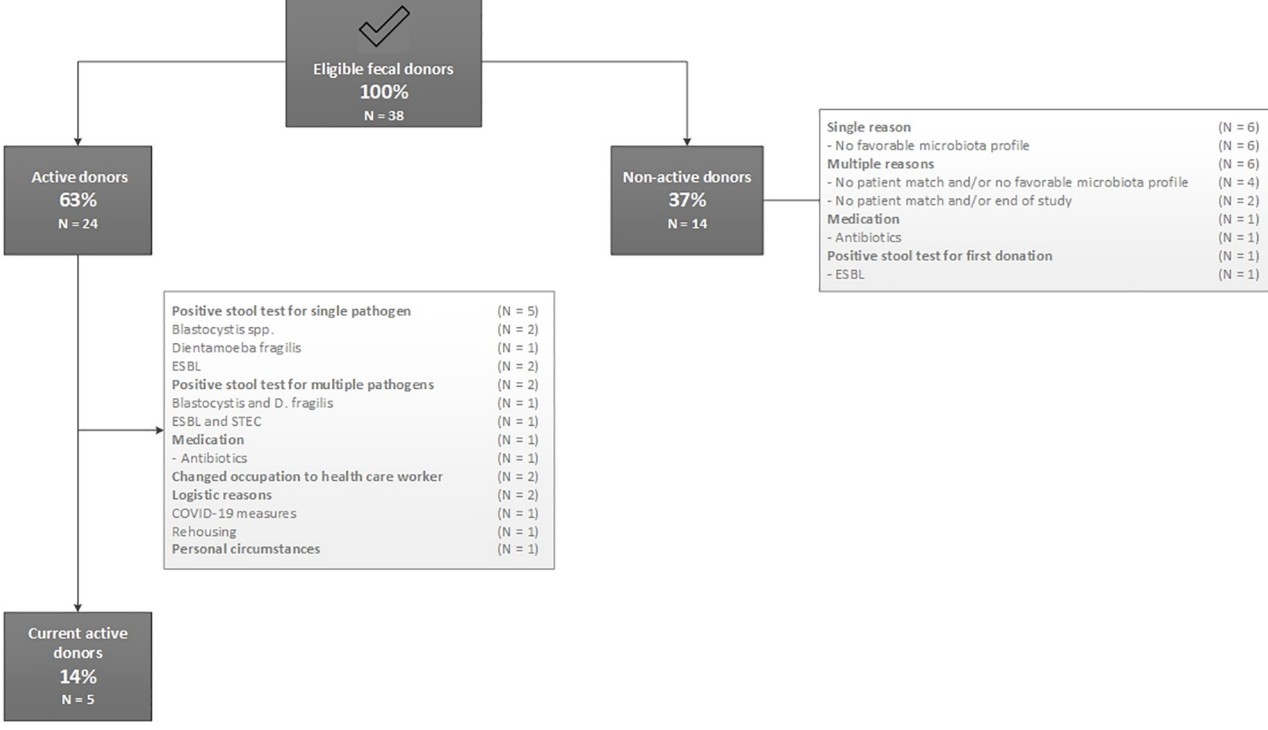

**Fig 2. Flow diagram of follow-up of eligible donors.**

not donate (demographic and referral reasons are listed in S1 Table). The number of donations per active donor ranged from 2 to 48 with a median of twelve donations (IQR: 5.3–18.8). Seven donors donated for and participated in multiple studies. The active donors (N = 24) had a median participation time of 13 months (IQR: 8–16 months). Additional screenings due to symptoms or exposure to risk factors were performed in 11 donors with a total of 34 tests, of which 11 (32.6%) returned positive. Five donors had transient positive tests that didn't lead to definite exclusion, most frequently a transient infection with enterovirus. Reasons for definite exclusion of active donors varied; six donors were excluded due to recurrent positive stool testing of which the majority tested positive for *Dientamoeba fragilis* and/or microscopic quantification of 'moderate' or 'many' *Blastocystis spp.* (N = 4). Demographic characteristics of the active donors, details on (re)screenings, and reasons for later exclusion are listed in S2 Table. At time of writing (August 2021), only five out of the 38 eligible donors (13%) were still qualified and active donors. The median time of their participation up till August 2021 was 9 months (IQR: 4–21.5 months).

## Screening costs

An initial safety screening at our center amounted to €846 for all fecal and blood tests only, not including microbiota profiling (TURN2), costs for location, travel allowance and compensation for donors, and wage of study coordinators (Table 3). The total cost of all performed biochemical tests was €64,112 to find 38 eligible fecal donors. Total screening costs per active donor were estimated at €2,774 a year, including full initial screening, one full rescreening (every six months), four times an additional 60-day screening, and average costs of additional

**Table 3. Total costs donor screening procedure.**

| Screening | € |
|---|---|
| **Full screening** | **846** |
| *Feces screening* | *657* |
| *Serum screening* | *189* |
| **60—day screening** | **195** |
| *Multidrug resistant organisms* | *150* |
| *SARS-CoV-2* | *45* |
| **Full rescreening 4 months (*TURN2*)** | **940** |
| *Feces screening* | *612* |
| *Serum screening*[a] | *328* |
| **Full rescreening 6 months (*FAIS, IMITHOT, PIMMS*)** | **846** |
| *Feces screening* | *657* |
| *Serum screening* | *189* |
| **Additional rescreening**[b] | **Variable** |
| *Feces bacteria* | *150* |
| *Feces MDRO* | *150* |
| *Feces viral gastroenteritis* | *45* |
| *Feces SARS-CoV-2* | *45* |
| *Feces parasites* | *212* |
| *Serum hematology* | *44* |

[a] Full rescreening in TURN2-trial included PCR assays of HIV, CMV, HBV and HCV instead of serology;

[b] In case of concerns donation was suspended and an additional rescreening was performed depending on symptoms and/or exposed risk factor of the fecal donor.

screenings per active donor (€197,-). In the TURN2-trial, in which frozen feces is used, the total screening costs per year are even higher; €5,388 a year per active donor, including full initial screening, two complete re-screenings (every four months) with PCR assays, three additional 60-day screening, and average costs of additional screenings per active donor (€197,-).

## Discussion

The expanding use of FMT in daily practice and clinical trials is accompanied by a need for more long-term available fecal donors and feasible donor screening programs. In this study, we reported our experience with stool donor recruitment, screening, follow-up, and associated costs in the context of clinical FMT trials. Our study showed that only 10% of potential donors passed all screening steps and could be enrolled as stool donors. Adding to the current literature, we reported the follow-up of eligible donors. In our experience, once qualified, active donors were eligible to donate for about a year before exclusion. Recruiting eligible donors is not only challenging, but also costly; we spent over €64,000 on biochemical tests only to detect 38 suitable fecal donors. This study highlights the obstacles in donor screening and provides practical insights for FMT researchers.

Previous research on donor screening showed variable success rates between 0.8–31% [15–21]. Our 10% eligibility rate is similar to smaller studies performed by Craven et al. [15] and Paramsothy et al. [19]. A higher success rate compared to our data was reported in a Danish study, and may be explained by the fact that in this study potential donors were recruited among an existing cohort of eligible blood donors, in which the risk of transmittable infectious diseases by blood transfusion is already assessed [21]. Lower success rates were published by Openbiome, the first public stool bank based in the USA, in which over 15.000 candidates were (pre-)screened and only 3% eventually qualified as fecal donors. The majority of candidates (66%) failed prescreening mainly due to social history reasons and body-mass index higher than 30 kg/m$^2$ [27]. In our cohort approximately half of all potential donors failed prescreening (N = 198) of which half was lost to follow-up after initial contact (N = 97). Especially during the COVID-19 pandemic, when in periods employees were requested to work from home in accordance to national measures, we experienced high rates of exclusion due to logistics of stool donations. It could be assumed that the COVID-19 pandemic also impacted our high rates of lost to follow-up during prescreening. More insight into motivation and preferences around stool donation is needed to improve initial donor recruitment and to reduce drop-out rates. Limited data on this subject is available [28]. Based on a multinational questionnaire study, McSweeney and colleagues identified that a male gender and being a blood donor is associated with a high willingness to stool donation, whereas a lack of knowledge on FMT and logistic barriers associated with time and transportation around screening and stool donation were reported as deterrents [28]. These variables should be taken into consideration. In general, the process of screening and donating should be as easy and convenient for donors as possible.

The global distribution of donor exclusion reasons varies not only as a result of different screening criteria between FMT centers and stool banks [9] but also on diagnostic approach and geographic location. For example, in the Hong Kong study stool tests were failed by the majority (86%) due to the carriage of ESBL-producing *Enterobacteriaceae* [16]. High prevalence of ESBL in this area is the result of several factors, including a high population density and diet habits. High carrier ship of ESBL-producing organisms seems less of an issue for donor selection in the USA and the Netherlands, where stool bank Openbiome tested only 3 of 571 (0.5%) stool donors positive for MDROs [27], and in our experience, ESBL positive stool tests accounted for the exclusion of five (8.8%) Dutch individuals at initial screening.

Moreover, the US FDA has warranted screening for enteropathogenic E. coli (EPEC) by stool nucleic acid amplification testing (NAAT) in addition to Shiga toxin-producing E. coli (STEC) [29]. In our cohort one individual failed stool testing due to the presence of STEC. Currently, EPEC is not included in our screening, because data on pathogenicity of EPEC is inconclusive [30]. Including EPEC in our screening protocol could result in even higher rates of donor exclusion.

In our cohort, we found positive parasite testing as the most common exclusion reason during the laboratory screening stage (91 out of 148 stool samples, 61.5%), in specific the presence of *Dientamoeba fragilis* or high amounts of *Blastocystis* spp. This is why, at least in certain cohorts, parasitology testing should follow as first step of the laboratory testing phase after (pre-)screening questionnaires. *D.fragilis* and *Blastocystis* spp. were also leading reasons for exclusion in the Canadian study by Craven et al. [15] and the Australian study by Paramsothy et al. [19], but not in others [16, 17]. Despite the recommendation of an international guideline to screen and exclude for these protozoa, heterogeneity between screening procedures in practice exists. According to a systematic review evaluating 168 FMT studies, only 15.7% and 14.5% of studies specifically report screening for *D. fragilis* and *Blastocystis* spp., respectively [9]. Moreover, many studies do not state the methods to screen for these organisms, even though the specific diagnostics used has a considerable influence on the detection rate. To illustrate this, the introduction of a *Blastocystis spp.* polymerase chain reaction (PCR) test by the NDFB in 2018 resulted in the discovery that feces from previously by-microscopy-regarded *Blastocystis* spp.-negative donors did actually contain DNA of *Blastocystis subtypes 1* or *3* and that these *Blastocystis* spp. were transferred to 31 patients via FMT [31]. Importantly, this did not have a negative effect on the efficacy of treatment for CDI nor resulted in gastrointestinal symptoms. The potential risk of harming recipients by transferring *Blastocystis* spp., might be overestimated. In fact, patients that received *Blastocystis* spp.-positive donor stool evaluated their defecation pattern in the long-term as more improved than those receiving *Blastocystis spp.*-negative donor stool [31].

Current consensus recommendations for screening stool donors are based on safety criteria, drawn up by FMT experts in the field, and aim to minimize the risk of inadvertently transmitting a communicable disease to an FMT recipient. Once a potential pathogen is added to the screening norms it can be difficult to defer it later. However, since the field of FMT research is still relatively new, these criteria are not always supported by solid data and should therefore be adjusted to risk-benefit analysis and progressive insights. For example, whether the exclusion of *D. fragilis*- and *Blastocystis* spp.-positive donors is justified could be questioned, especially for *Blastocystis* spp. of which the pathogenicity is still under debate [32, 33]. Both *Blastocystis* spp. and D. *fragilis* appear more commonly in asymptomatic individuals than in patients with gastrointestinal symptoms or disorders, suggesting that these protozoa can have a commensal relationship with human hosts [34–36]. Interestingly, recent literature shows a link between the presence of the above-mentioned single-cell eukaryotes, especially *Blastocystis* spp., and gut microbiota features [37]. For example, stool containing *Blastocystis* spp. has been associated to higher bacterial diversity and distinct microbial profiles (*e.g.* enterotype Bacteroides [38] and co-occurrence with the beneficial bacteria *Akkermansia* [39]), and their presence may reflect a healthier state of the gut microbiota [38–43]. The application of the current consensus screening protocol that suggests the exclusion of *Blastocystis* spp. positive donors [14] could therefore result in the elimination of stool donors that have a favorable bacterial community. This led us to adjust our initial screening protocols where we now accept donors with microscopic quantification of 'rare' or 'few' *Blastocystis* spp. and only exclude individuals with 'moderate' of 'many'*Blastocystis* spp. [26]. Due to the double-blinded nature of the described ongoing clinical studies, it is not yet established if *Blastocystis* spp. positive

FMT products have been transferred to our study patients. To prevent unnecessary elimination of valuable stool donors, future research should look into the influence of co-transplantation of common protozoa (and their subtypes) on the microbiota structure and efficacy of FMT.

Since there is limited understanding of what constitutes a successful stool donor for different conditions, most current screening protocols do not comprise potential predictors for FMT efficacy. Nevertheless, it is clear that FMT can improve disease outcome in some recipients (responders), but not in all (non-responders). Hence, the current 'one stool fits all' approach may not be the way to go [44–46]. A more personalized donor-recipient matching strategy where donors are screened for taxa associated with metabolic pathways, or directly for metabolites [47], that are disturbed in a particular disease phenotype, might enhance FMT efficacy. Conversely, the more tailor-made matching strategies will become, the harder the search for suitable donors will be. Evidently, future larger-scale studies in the FMT field are needed to further explore donor-dependent predictors of treatment success.

In the current trials, 14 eligible donors could not be matched to recipients based on gender and/or CMV/EBV status. These mismatches led to expiration of costly screening results and non-activity of valuable stool donors. This waste of screening costs is partly explained by the fact that the current trials started with establishing a pool of healthy stool donors, whereas at that time no patients were included and stool donation was not yet required. Donor-recipient incompatibilities could be prevented by a more synchronous approach of execution of donor screening programs and patient recruitment. Alternatively, especially in trials using fresh fecal material for FMT, another approach could be applied where patients are first recruited and serologically profiled and subsequently a suitable donor is being sought. The stepwise approach for donor screening could then start with serological testing for pathogenic viruses. Only in case of gender and/or CMV/EBV match, the potential donor could continue full screening program. However, postponing the search of stool donors until a study patient has been screened, might result in an unnecessary delay in the start of study treatment.

Direct costs of an initial safety screening at our center was €846 (891 USD) per donor. These costs did not include overhead, administration costs and personnel. Limited data is available on associated donor screening costs in other centers. In accordance with our study, Kazerouni et al. [48]. evaluated screening costs for Openbiome to be 885 USD per donor, including clinical assessment, stool and serum screening. The Canadian study by Craven et al. [15] reported that the costs for a full donor screening work-up (including history, examination, blood, stool, and urine screening, and administration) were approximately 440 USD per donor. Differences in costs can be explained by lower costs of biochemical tests in Canada. As discussed previously, minor differences in screening protocols occur since no current consensus on the perfect screening program exists. Furthermore, since there is no global consensus on the classification of human stool used for FMT (e.g. medicinal product, biological product, human cell/tissue product, or substance of human origin) differences in associated codes and regulations between countries exist [49, 50]. It should be considered that stricter regulations can lead to increased rates of (temporarily) donor disqualification and even higher associated donor screening costs. Examples of stricter regulations compared to our donor protocol are more regular rescreening of active fecal donors, screening for more enteric pathogens (e.g. EPEC implemented by OpenBiome [51]), broader assessment of conditions (e.g. anti-nuclear antibody test for autoimmune diseases [52]), and mandatory donation of feces in a dedicated supervised bathroom (resulting in higher logistic, facility and employee expenses). By reporting the average costs associated with our donor screening program we provide an estimate for clinicians thinking of establishing a pool of healthy stool donors for FMT research. Collaboration with other FMT researchers or national stool banks, in order to share screening costs and

eligible donors, will presumably be more cost-effective. Furthermore it lowers the chance of discarding valuable FMT products when a suitable patient match cannot be found within a relatively small study cohort.

Nowadays, FMT is a widely accepted treatment for recurrent *Clostridioides difficile* infection [1, 2]. The application of FMT as a treatment for other conditions associated with alterations in the gut microbiome, is limited to the context of clinical trials [8–10]. This barrier has driven some patients to seek for alternative options, including Do-It-Yourself-FMT procedures with self-administration of (mostly) unscreened donor feces [53]. The high rates of donor exclusions in seemingly healthy individuals reported by our study and other FMT programs [15–21] illustrates that Do-It-Yourself-FMT procedures can be accompanied by several risks, most importantly the risk of inadvertent transmission of a infectious disease to an FMT recipient. Ekekezie et al. studied factors that influenced willingness to pursue DIY-FMT. Results showed that majority of respondents would have preferred to have FMT performed in a clinical setting [53]. However, lack of access drives these patients to try FMT at home. Regulated stool banks could partially attenuate this problem by enabling compassionate use of FMT in carefully defined clinical cases. A major advantage of regulated (national) stool banks is to ensure safety of FMT products by following strict safety criteria for screening stool donors. Nevertheless, health care professionals must acknowledge the fact that DIY-FMT is an actual phenomenon and therefore clinicians should discuss concerns regarding safety and potential harms with patients considering such a procedure. On the other hand, commercial developers argue that the development of synthetic microbial community products seem to be a safe and sustainable alternative to conventional FMT [54]. However, most colonic bacteria are yet unculturable not and current synthetic microbial products contain limited strains and therefore poorly represent the gut microbiome. Data on clinical efficacy of these products as well as their longterm safety is yet unavailable. Also, data on transmission of uncovered harmful species (i.e. potentially procarcinogenic or pathogenic) can only be derived retrospectively from performed conventional FMT studies [55]. Using synthetic microbial products in FMT trials would rule out the possibilities for these ancillary findings.

This study has several strengths. Firstly, our study included data regarding recruitment and selection procedures of healthy fecal donors from four different clinical FMT trials, creating a large cohort. Secondly, by presenting follow-up data we provided information on the time frame in which donors were qualified to donate feces after successful screening. Furthermore, this study included an estimation of donor screening costs. By presenting discussed data, this study provides insights in the challenges for creating a sustainable feces donor pool and is accordingly relevant for researchers setting up clinical FMT trials.

Nonetheless, this study also has some limitations. First, the FMT trials required donors to deliver fresh fecal samples to the hospital for rapid procurement. Therefore, only donors living within a short travel distance were included, comprising mostly urban areas. This potentially influenced the presence of pathogenic microorganisms as mentioned above and limits the generalizability of our results to other regions and countries. Secondly, due to our stepwise screening approach not all fecal and blood laboratory tests were executed on every potential donor. Therefore, presented data on donor deferral reasons per step should be interpreted with caution. Lastly, as discussed, minor differences in the screening protocols of the four included clinical trials were present. Pre-screening approaches through advertising and short telephonic interviews to discuss in- and exclusion criteria were not standardized. As a consequence, possible exclusions of potential donors and multiple donor deferral reasons could have been missed. Nevertheless, the most relevant in- and exclusion criteria were similar and our approach is in line with current available screenings protocols [14, 23]. Therefore, we believe that the effect of the minor (pre-) screening differences is limited.

## Conclusion

In conclusion, this study shows that a thorough screening protocol for stool donors in the context of clinical FMT trials results in only 10% being eligible donors and is associated with substantial costs. The majority of healthy asymptomatic donors failed stool testing, predominately due to positive parasite testing. The need to exclude donors that carry certain protozoa, especially *Blastocystis* spp., is questionable. The high rates of donor exclusions in seemingly healthy individuals reported by our study illustrates that Do-It-Yourself-FMT procedures can be accompanied by several risks, Further research into the centralization of stool donor screening and procurement of FMT products is warranted.

## Supporting information

**S1 Table. Demographics and reasons of exclusion of non-active donors.** [a] based on gender and/or CMV/EBV status; [b] PIMMS or FAIS study; [c] Donors of the TURN2-trial were additionally selected on a putatively favorable microbiota profile based on results from a previous TURN1 trial. Abbreviations: ESBL, extended spectrum beta-lactamase.
(DOCX)

**S2 Table. Demographics, specifications of screening, and reasons of exclusion of active donors.** [a] Determined microscopically by an experienced laboratory analyst [26]; [b] PIMMS or FAIS study; [c] based on gender or CMV/EBV status. Abbreviations: CBC, complete blood count; CRP, c-reactive protein; DFT, dual feces test; ESBL, extended spectrum beta-lactamase; NA, not applicable; STEC, shiga toxin-producing Escherichia coli; MDROs, multidrug resistant organisms; SARS-CoV-2, severe acute respiratory syndrome coronavirus 2.
(DOCX)

## Acknowledgments

The authors are grateful to all the potential stool donors who participated in the study. We thank research analyst Patricia Broekhuizen for the work she performed around coordinating and analysing the Dual Feces Tests. We thank Djuna de Jong for her support on screening donors. Lastly we thank Anouschka Komproe for assistance in data processing.

## Author Contributions

**Conceptualization:** Mèlanie V. Bénard, Clara M. A. de Bruijn, Aline C. Fenneman, Koen Wortelboer, Judith Zeevenhoven, Bente Rethans, Hilde J. Herrema, Max Nieuwdorp, Marc A. Benninga, Cyriel Y. Ponsioen.

**Data curation:** Mèlanie V. Bénard, Clara M. A. de Bruijn, Aline C. Fenneman, Koen Wortelboer, Judith Zeevenhoven, Bente Rethans.

**Formal analysis:** Mèlanie V. Bénard, Clara M. A. de Bruijn.

**Funding acquisition:** Max Nieuwdorp, Marc A. Benninga, Cyriel Y. Ponsioen.

**Investigation:** Hilde J. Herrema, Max Nieuwdorp, Marc A. Benninga, Cyriel Y. Ponsioen.

**Methodology:** Mèlanie V. Bénard, Clara M. A. de Bruijn.

**Project administration:** Mèlanie V. Bénard, Clara M. A. de Bruijn, Aline C. Fenneman, Koen Wortelboer.

**Supervision:** Hilde J. Herrema, Tom van Gool, Max Nieuwdorp, Marc A. Benninga, Cyriel Y. Ponsioen.

**Writing – original draft:** Mèlanie V. Bénard, Clara M. A. de Bruijn.

**Writing – review & editing:** Aline C. Fenneman, Koen Wortelboer, Judith Zeevenhoven, Bente Rethans, Hilde J. Herrema, Tom van Gool, Max Nieuwdorp, Marc A. Benninga, Cyriel Y. Ponsioen.

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
