## [Decision Letter · Decision Letter 0]

18 Jul 2022

PONE-D-22-15731Challenges and costs of donor screening for fecal microbiota transplantationsPLOS ONE

Dear Dr. Bénard,

Thank you for submitting your manuscript to PLOS ONE. After careful consideration, we feel that it has merit but does not fully meet PLOS ONE’s publication criteria as it currently stands. Therefore, we invite you to submit a revised version of the manuscript that addresses the points raised during the review process.

Based on the reviewers' comments, there is one important question the authors should consider: Is PloS One the appropriate outlet for such a manuscript. Reviewer 1 who is the statistics expert and reviewer 2 to a lesser extent point issues related to sample size. Those are both inherent and part of the "challenges" announced by the title. I would say that the manuscript is publishable after addressing Reviewer 2 and 3 comments, but I strongly encourage the authors to contact the PloS technical editors to ask about the suitability of such a "perspective/editorial" piece in the journal. 

We look forward to receiving your revised manuscript.

Kind regards,

Franck Carbonero, PhD

Academic Editor

PLOS ONE

Journal Requirements:

"MN is supported by a ZONMW VICI grant 2020 [09150182010020]. There were no other specific grant for this research from any funding agency in the public, commercial or not-for-profit sectors."

Additional Editor Comments:

Based on the reviewers' comments, there is one important question the authors should consider: Is PloS One the appropriate outlet for such a manuscript. Reviewer 1 who is the statistics expert and reviewer 2 to a lesser extent point issues related to sample size. Those are both inherent and part of the "challenges" announced by the title. I would say that the manuscript is publishable after addressing Reviewer 2 and 3 comments, but I strongly encourage the authors to contact the PloS technical editors to ask about the suitability of such a "perspective/editorial" piece in the journal.

Reviewers' comments:

Reviewer's Responses to Questions

**Comments to the Author**

1. Is the manuscript technically sound, and do the data support the conclusions?

Reviewer #1: No

Reviewer #2: Yes

Reviewer #3: Yes

2. Has the statistical analysis been performed appropriately and rigorously? 

Reviewer #1: No

Reviewer #2: N/A

Reviewer #3: Yes

3. Have the authors made all data underlying the findings in their manuscript fully available?

Reviewer #1: Yes

Reviewer #2: Yes

Reviewer #3: Yes

4. Is the manuscript presented in an intelligible fashion and written in standard English?

Reviewer #1: Yes

Reviewer #2: Yes

Reviewer #3: Yes

5. Review Comments to the Author

Reviewer #1: The statistical approach was primarily descriptive as there are no real comparisons being made. The overall 10% participation rate (38/393) is not very convincing from a statistical perspective. How relevant or generalizable the statistics presented are to a realistic screening cost attempt is doubtful. The investigators made no attempt to determine the true precision ( using appropriate statistical sampling techniques) of any subject data based on this small sample.

Another deficit is that there was no effort to compare the subject characteristics seen in the various tables and supplemental tables in the active and non-active donor groups.

The manuscript has to be edited for typos. On line 227 of the text the investigators state that,” Demographic characteristics of the active donors, details on (re)screenings, and reasons for later exclusion are listed in Supplementary Table S2". The active donor data is on Table S1.

Reviewer #2: Benard et al. describe their experience in qualifying and sustaining stool donors for FMT trials for IBS, autoimmunity, obesity, and ulcerative colitis. They quantify the donor-associated costs within their specific protocol. Reporting this experience is valuable for several audiences: academic centers thinking about establishing their own program, commercial developers of microbiota-based therapeutics, and regulators.

Perceptions of how easy or hard it is to run a donor program are varied and contradictory. Some assume that such a program should be easy and inexpensive because stool is widely available. Others believe that procuring stool as raw material for therapeutics is fraught with insurmountable dangers and will ultimately be looked upon as a transient period in medical history forced by desperation.

It is important to note that this is still a very young field. Little is known about the dangers lurking in the stool of asymptomatic and seemingly healthy individuals. Experts often base their opinions on minimal evidence, which can be entirely hypothetical or anecdotal. There is often a somewhat flippant attitude to add to the lists of screening tests, which can then be accepted by regulators as settled knowledge and written in stone without adequate evaluation of true risks and benefits. There is the other side, common among many patients and do-it-yourself advocates, that FMT is entirely innocuous and free of dangers. These attitudes should be carefully considered in this manuscript.

Overall, I think the manuscript is very well written, and the following are my comments that I hope will be addressed:

1. The exclusion rate of donors based on findings of D. fragilis and Blastocystis spp. is striking. However, as the authors note, these ‘parasites’ are more likely to be innocuous commensals and may even be mutualist as the carriers may have lesser prevalence of GI symptoms and their microbiota may have more favorable characteristics (e.g., bacterial diversity, butyrate production). I think the modified protocol presented in the discussion is reasonable, although difficult to implement given subjectivity intrinsic to ‘high’ or ‘low’ abundance as determined by a laboratory technician. An RCT with D. fragilis/Blastocystis-positive versus -negative FMT may be the best way to resolve this question, especially in the context of IBS-focused clinical trials. I recommend that the questionable value of inclusion of testing for these ‘parasites’ into the donor screening protocol should be somehow included in the abstract, given that this might be the only part of the paper read by many.

2. While the donor screening and testing protocol described is reasonably rigorous, I think the discussion should consider additional costs that they could be facing if they had to adhere to even stricter regulations:

a. One potential pathogen that is already written in stone in the US by the FDA is enteropathogenic E. Coli (EPEC), which is further mandated to be tested for using nucleic acid amplification. EPEC is commonly found in healthy individuals, may a self-limited diarrheal illness, although little is known about the pathogenicity of its various strains. If it were included in the screening protocol of this group, they would have an even higher rate of exclusion. Absence of EPEC testing should be noted in the discussion, as it is a serious burden for the donor programs in the US (and could become so in other countries as the regulators look to each other for else they should be doing).

b. The investigators describe the logistic difficulties in procuring stool during the COVID era because their donors had to bring in donations from home. While COVID has certainly complicated operations of stool donor programs, consider that the FDA requires US donors to produce stool on site in a supervised bathroom to ensure chain of custody.

c. The enteric pathogen testing protocol uses 60-day brackets. However, the rate of transiently positive tests would likely be higher if every donation was required to be tested for all pathogens. The added costs obviously include the additional labs expenses. However, flickering positive results lead to increased rates of donor disqualification.

d. Excessive testing may not be limited to enteric pathogens. For example, some FMT donor protocols in the US require anti-nuclear antibody testing, as a potential biomarker of future autoimmunity risk.

3. The authors should discuss their approach to discussing meaningless positive tests with their donors. Are donors notified of their test results and counseled? How do they deal with the potential anxiety that is then generated? The donors may think that they have some problem brewing or that they could transmit an infection to someone around them. If they do not have this discussion, why not?

4. The authors state in the introduction that stool testing for SARS-CoV-2 is required. Unfortunately, in the US stool testing for SARS-CoV-2 is not FDA approved. However, frequent nasopharyngeal testing on donors is allowed in some programs. I suggest the authors remove ‘stool’ from their statement – the methods of testing for SARS-CoV-2 are not a universally settled issue at this time.

5. The discussion should make a point that high rates of donor exclusion in a typical FMT donor program illustrate the dangers of do-it-yourself (DIY) protocols and the advantages of a centralized ‘stool-bank’ model. Yet, flippant over-regulation can drive such models out of existence and encourage greater uptake of DIY practices and medical tourism.

On the other hand, some commercial developers and academics argue that FMTs should go out of existence and be replaced with synthetic microbial products or alcohol-treated microbiota. However, these assertions are made without data that include microbiome-based endpoints. FMT has been shown to decrease the burden of antibiotic resistance genes in patients with recurrent C. difficile infections. Similarly, Sam Nooji et al. (Gastroenterology, 2021), showed that the prevalence of pks+ E. coli was either reduced or unchanged following FMT. How would synthetic products do with respect to these endpoints? I think the authors should anticipate these topics in their discussion because there is high likelihood that certain commercial players will capitalize on the ‘dangers’ of FMT suggested here (without evidence of harm in patients) and tout unvalidated advantages of their products.

Reviewer #3: Reviewer name: Jeremiah Faith

The manuscript by Bernard et.al., on the costs and screening challenges for FMT donors is an excellent resource that will be of interest for groups with their own FMT banks. Even more so, it is a great set of data and insights for laboratories/institutions looking to establish their own FMT banks. I have no major concerns. The major challenge with the read is that procedures very from bank to bank and regulations can vary across countries (and are not that specific anywhere to begin with). However this caveat is clearly stated and references to relevant additional reading are provided.

Minor comments:

1. I think the labels on top of supplemental Tables 1 and 2 are switched.

2. line 131 “existed” I think that should be “consisted” of non-smoking adults…

3. line 160 “alfa” should be “alpha”

4. It would be nice to hear more opinions from the authors one which of the tests they think should and should not be included (the D. fragilis and Blastocystis was helpful in this regard, more along these lines would be even better)

6. PLOS authors have the option to publish the peer review history of their article (what does this mean?). If published, this will include your full peer review and any attached files.

Reviewer #1: No

Reviewer #2: No

Reviewer #3: **Yes: **Jeremiah Faith

---

## [Author Response · Author response to Decision Letter 0]

5 Sep 2022

REVIEWERS COMMENTS

Reviewer: 1

Response to the reviewers’ queries: 

1. The statistical approach was primarily descriptive as there are no real comparisons being made. The overall 10% participation rate (38/393) is not very convincing from a statistical perspective. How relevant or generalizable the statistics presented are to a realistic screening cost attempt is doubtful. The investigators made no attempt to determine the true precision (using appropriate statistical sampling techniques) of any subject data based on this small sample.

With the current manuscript we aim to provide tools from a practical perspective to clinicians, investigators, as well as institutions who wish to set up or improve an FMT service for clinical use and/or research. Moreover, as our study included data regarding recruitment and selection procedures of healthy fecal donors from four different clinical FMT trials, we believe we created a relatively large cohort. By reporting the average costs associated with our donor screening program in our manuscript we provide an estimate for clinicians thinking of establishing a pool of healthy stool donors for FMT research. The generalizability of the presented screening costs are discussed on page 20-21. Due to the nature of the study aims, formal statistics are outside the scope of the manuscript.

2. Another deficit is that there was no effort to compare the subject characteristics seen in the various tables and supplemental tables in the active and non-active donor groups.

Main reason for donors to become non-active was the absence of a favorable microbiota profile (TURN2-trial). As described in the method section on page 7 a favorable microbiota profile is based on results from a previous TURN1 trial and include high alpha-diversity and high predicted butyrate production. Differences in subject characteristics with regard to the presence or absence of a favorable microbiota profile are outside the scope of the current study and will be published elsewhere within the context of the TURN2 trial. Hence, in this study, our aim was not to compare characteristics of active versus non-active donors.

3. The manuscript has to be edited for typos. On line 227 of the text the investigators state that,” Demographic characteristics of the active donors, details on (re)screenings, and reasons for later exclusion are listed in Supplementary Table S2". The active donor data is on Table S1.

We adjusted the titles of S1 Table and S2 accordingly.

Reviewer: 2

Benard et al. describe their experience in qualifying and sustaining stool donors for FMT trials for IBS, autoimmunity, obesity, and ulcerative colitis. They quantify the donor-associated costs within their specific protocol. Reporting this experience is valuable for several audiences: academic centers thinking about establishing their own program, commercial developers of microbiota-based therapeutics, and regulators.

Perceptions of how easy or hard it is to run a donor program are varied and contradictory. Some assume that such a program should be easy and inexpensive because stool is widely available. Others believe that procuring stool as raw material for therapeutics is fraught with insurmountable dangers and will ultimately be looked upon as a transient period in medical history forced by desperation.

It is important to note that this is still a very young field. Little is known about the dangers lurking in the stool of asymptomatic and seemingly healthy individuals. Experts often base their opinions on minimal evidence, which can be entirely hypothetical or anecdotal. There is often a somewhat flippant attitude to add to the lists of screening tests, which can then be accepted by regulators as settled knowledge and written in stone without adequate evaluation of true risks and benefits. There is the other side, common among many patients and do-it-yourself advocates, that FMT is entirely innocuous and free of dangers. These attitudes should be carefully considered in this manuscript.

Overall, I think the manuscript is very well written, and the following are my comments that I hope will be addressed:

Response to the reviewers’ queries: 

1. The exclusion rate of donors based on findings of D. fragilis and Blastocystis spp. is striking. However, as the authors note, these ‘parasites’ are more likely to be innocuous commensals and may even be mutualist as the carriers may have lesser prevalence of GI symptoms and their microbiota may have more favorable characteristics (e.g., bacterial diversity, butyrate production). I think the modified protocol presented in the discussion is reasonable, although difficult to implement given subjectivity intrinsic to ‘high’ or ‘low’ abundance as determined by a laboratory technician. An RCT with D. fragilis/Blastocystis-positive versus -negative FMT may be the best way to resolve this question, especially in the context of IBS-focused clinical trials. I recommend that the questionable value of inclusion of testing for these ‘parasites’ into the donor screening protocol should be somehow included in the abstract, given that this might be the only part of the paper read by many.

We thank the reviewer for this comment. We have discussed this matter among the co-authors and decided to add a sentence on this subject to the abstract. However, we made a distinction between protozoa Dientamoeba Fragilis and Blastocystis spp. on this statement after elaborate discussion with the parasitology expert of our center, Dr. T. van Gool. Data on pathogenicity of Dientamoeba Fragilis is widely available and several international guidelines advise to treat carriership with Dientamoeba when gastro-intestinal symptoms are present and no other cause is found. To advice to stop screening for Dientamoeba in donor feces is in our opinion not supported by (our) data. 

2. While the donor screening and testing protocol described is reasonably rigorous, I think the discussion should consider additional costs that they could be facing if they had to adhere to even stricter regulations:

a. One potential pathogen that is already written in stone in the US by the FDA is enteropathogenic E. Coli (EPEC), which is further mandated to be tested for using nucleic acid amplification. EPEC is commonly found in healthy individuals, may a self-limited diarrheal illness, although little is known about the pathogenicity of its various strains. If it were included in the screening protocol of this group, they would have an even higher rate of exclusion. Absence of EPEC testing should be noted in the discussion, as it is a serious burden for the donor programs in the US (and could become so in other countries as the regulators look to each other for else they should be doing).

We thank the reviewer for raising this matter. We have added the difference with regard to screening for EPEC between FMT centers in the discussion on page 18. We have now highlighted that if EPEC was included in our screening protocol, this would result in even higher rates of donor exclusion. Moreover, we elaborate on this matter in the discussion on page 21 by focusing on potential rising donor screening costs. 

b. The investigators describe the logistic difficulties in procuring stool during the COVID era because their donors had to bring in donations from home. While COVID has certainly complicated operations of stool donor programs, consider that the FDA requires US donors to produce stool on site in a supervised bathroom to ensure chain of custody.

We agree with the reviewer that we could have elaborated a bit more on the method of sample collection. We adjusted the methods of stool sample collection on page 7. We believe that by using universal clean plastic containers for collecting stool the risk of transmitting communicable disease is negligible. We discuss the potential of rising donor screening costs when stool production in a supervised bathroom becomes mandatory in the discussion on page 21. 

c. The enteric pathogen testing protocol uses 60-day brackets. However, the rate of transiently positive tests would likely be higher if every donation was required to be tested for all pathogens. The added costs obviously include the additional labs expenses. However, flickering positive results lead to increased rates of donor disqualification.

We thank the reviewer for this suggestion and adjusted the discussion on page 18 accordingly.

d. Excessive testing may not be limited to enteric pathogens. For example, some FMT donor protocols in the US require anti-nuclear antibody testing, as a potential biomarker of future autoimmunity risk.

We thank the reviewer for this remark and added testing for anti-nuclear antibodies on page 20 as an example of differences (or possible future adaptations) to our screening protocol that will increase screening costs 

3. The authors should discuss their approach to discussing meaningless positive tests with their donors. Are donors notified of their test results and counseled? How do they deal with the potential anxiety that is then generated? The donors may think that they have some problem brewing or that they could transmit an infection to someone around them. If they do not have this discussion, why not?

We thank the reviewer for these relevant questions. Our IRB approved protocols stipulate to notify donors of any positive test results and counsel them accordingly. In our experience and to the best of our knowledge, these counselling sessions did not lead to anxiety in our potential donors. We also discuss the chance of finding clinically irrelevant test results beforehand in the informed consent progress. 

In the manuscript (in the Methods section, sentence 165,166,167 we added our approach with discussing positive test values with our donors. 

4. The authors state in the introduction that stool testing for SARS-CoV-2 is required. Unfortunately, in the US stool testing for SARS-CoV-2 is not FDA approved. However, frequent nasopharyngeal testing on donors is allowed in some programs. I suggest the authors remove ‘stool’ from their statement – the methods of testing for SARS-CoV-2 are not a universally settled issue at this time.

We have removed ‘stool’ in the statement in the introduction on page 4.

5. The discussion should make a point that high rates of donor exclusion in a typical FMT donor program illustrate the dangers of do-it-yourself (DIY) protocols and the advantages of a centralized ‘stool-bank’ model. Yet, flippant over-regulation can drive such models out of existence and encourage greater uptake of DIY practices and medical tourism.

We agree with the reviewer that we should make a statement on the dangers of do-it-yourself FMT. We added this to the Discussion on page 20-21 and Conclusion.

6. On the other hand, some commercial developers and academics argue that FMTs should go out of existence and be replaced with synthetic microbial products or alcohol-treated microbiota. However, these assertions are made without data that include microbiome-based endpoints. FMT has been shown to decrease the burden of antibiotic resistance genes in patients with recurrent C. difficile infections. Similarly, Sam Nooji et al. (Gastroenterology, 2021), showed that the prevalence of pks+ E. coli was either reduced or unchanged following FMT. How would synthetic products do with respect to these endpoints? I think the authors should anticipate these topics in their discussion because there is high likelihood that certain commercial players will capitalize on the ‘dangers’ of FMT suggested here (without evidence of harm in patients) and tout unvalidated advantages of their products.

We thank the reviewer for sharing his/her concern that there is a likelihood that commercial developers will capitalize on the dangers of conventional FMT, derived from our presented data. We added the development of synthetic microbial products in the discussion on page 22. We highlight the limitations of the use of synthetic microbial products, namely: 1) current synthetic microbial products comprised of limited strains poorly represent the complete gut microbiome and 2) data on transmission of uncovered potentially procarcinogenic or pathogenic species cannot be derived retrospectively from trials using synthetic products, 3) the absence of clinical data on the efficacy and long-term safety of synthetic microbial products. 

Reviewer: Jeremiah Faith

The manuscript by Bernard et al., on the costs and screening challenges for FMT donors is an excellent resource that will be of interest for groups with their own FMT banks. Even more so, it is a great set of data and insights for laboratories/institutions looking to establish their own FMT banks. I have no major concerns. The major challenge with the read is that procedures very from bank to bank and regulations can vary across countries (and are not that specific anywhere to begin with). However this caveat is clearly stated and references to relevant additional reading are provided.

Response to the reviewers’ queries: 

1. I think the labels on top of supplemental Tables 1 and 2 are switched.

2. Line 131 “existed” I think that should be “consisted” of non-smoking adults…

3. Line 160 “alfa” should be “alpha”

These points have been adjusted in the manuscript.

4. It would be nice to hear more opinions from the authors one which of the tests they think should and should not be included (the D. fragilis and Blastocystis was helpful in this regard, more along these lines would be even better) 

 This comment is in line with the request of reviewer 2; we added our opinion with regard to screening for enteropathogenic E. coli (EPEC) discussion on page 18. To the best of our knowledge, there is no additional data to support the decision to include or exclude pathogens in our current screening protocol. Therefore, we did not further elaborate on this matter.

---

## [Decision Letter · Decision Letter 1]

21 Sep 2022

PONE-D-22-15731R1Challenges and costs of donor screening for fecal microbiota transplantationsPLOS ONE

Dear Dr. Bénard,

Thank you for submitting your manuscript to PLOS ONE. After careful consideration, we feel that it has merit but does not fully meet PLOS ONE’s publication criteria as it currently stands. Therefore, we invite you to submit a revised version of the manuscript that addresses the points raised during the review process.

Reviewer 2 has requested to see the edits and responses to his remaining minor comments.

We look forward to receiving your revised manuscript.

Kind regards,

Franck Carbonero, PhD

Academic Editor

PLOS ONE

Journal Requirements:

Reviewers' comments:

Reviewer's Responses to Questions

**Comments to the Author**

1. If the authors have adequately addressed your comments raised in a previous round of review and you feel that this manuscript is now acceptable for publication, you may indicate that here to bypass the “Comments to the Author” section, enter your conflict of interest statement in the “Confidential to Editor” section, and submit your "Accept" recommendation.

Reviewer #1: (No Response)

Reviewer #2: All comments have been addressed

Reviewer #3: All comments have been addressed

2. Is the manuscript technically sound, and do the data support the conclusions?

Reviewer #1: Partly

Reviewer #2: Yes

Reviewer #3: Yes

3. Has the statistical analysis been performed appropriately and rigorously? 

Reviewer #1: N/A

Reviewer #2: Yes

Reviewer #3: Yes

4. Have the authors made all data underlying the findings in their manuscript fully available?

Reviewer #1: Yes

Reviewer #2: Yes

Reviewer #3: Yes

5. Is the manuscript presented in an intelligible fashion and written in standard English?

Reviewer #1: Yes

Reviewer #2: Yes

Reviewer #3: Yes

6. Review Comments to the Author

Reviewer #1: There is nothing to evaluate here statistically. From that perspective, it is not very exciting or thorough.

I defer to the editors as per the final decision of suitabllity for publication.

Reviewer #2: The authors have provided thoughtful answers to the critiques and have done an admirable job overall. I have two remaining comments, which need to be addressed:

1. The authors said they addressed the issue of supervised bathroom donations on page 21. I saw a table on page 21 with no discussion of this issue. A supervised bathroom is mandatory in manufacturing of FMT products in FDA-approved protocols. The cost should be acknowledged, including (1) logistics of recruitment of donors required to come to a facility and provide stool on demand; (2) facility fees for reserving a dedicated bathroom; (3) having a donor coordinator on site to receive the donations.

2. I did not see a mention of whether the donors are paid or not, and how much. This is an important issue. Commercial manufacturers of FMT-like products, e.g., Seres Health, Ferring Pharmaceuticals (Rebiotix) pay quite handsomely to their donors, e.g., ≥ $1,200 per month. This practice obviously introduces potential hazards, which are amplified hugely by not having a supervised bathroom.

Reviewer #3: all comments responded to sufficiently.

7. PLOS authors have the option to publish the peer review history of their article (what does this mean?). If published, this will include your full peer review and any attached files.

Reviewer #1: No

Reviewer #2: No

Reviewer #3: **Yes: **Jeremiah Faith

---

## [Author Response · Author response to Decision Letter 1]

29 Sep 2022

Comments processed in the previous manuscript revision

---

## [Decision Letter · Decision Letter 2]

5 Oct 2022

Challenges and costs of donor screening for fecal microbiota transplantations

PONE-D-22-15731R2

Dear Dr. Bénard,

We’re pleased to inform you that your manuscript has been judged scientifically suitable for publication and will be formally accepted for publication once it meets all outstanding technical requirements.

Kind regards,

Franck Carbonero, PhD

Academic Editor

PLOS ONE

Additional Editor Comments (optional):

Reviewers' comments:

Reviewer's Responses to Questions

**Comments to the Author**

1. If the authors have adequately addressed your comments raised in a previous round of review and you feel that this manuscript is now acceptable for publication, you may indicate that here to bypass the “Comments to the Author” section, enter your conflict of interest statement in the “Confidential to Editor” section, and submit your "Accept" recommendation.

Reviewer #2: All comments have been addressed

2. Is the manuscript technically sound, and do the data support the conclusions?

Reviewer #2: Yes

3. Has the statistical analysis been performed appropriately and rigorously? 

Reviewer #2: N/A

4. Have the authors made all data underlying the findings in their manuscript fully available?

Reviewer #2: Yes

5. Is the manuscript presented in an intelligible fashion and written in standard English?

Reviewer #2: Yes

6. Review Comments to the Author

Reviewer #2: (No Response)

7. PLOS authors have the option to publish the peer review history of their article (what does this mean?). If published, this will include your full peer review and any attached files.

Reviewer #2: No

---

## [Editor Report · Acceptance letter]

12 Oct 2022

PONE-D-22-15731R2 

Challenges and costs of donor screening for fecal microbiota transplantations 

Dear Dr. Bénard:

I'm pleased to inform you that your manuscript has been deemed suitable for publication in PLOS ONE. Congratulations! Your manuscript is now with our production department. 

Kind regards, 

on behalf of

Dr. Franck Carbonero 

Academic Editor

PLOS ONE